# DexPoint: Generalizable Point Cloud Reinforcement Learning for Sim-to-Real Dexterous Manipulation

**Yuzhe Qin\*[1]**     **Binghao Huang\*[1]**     **Zhao-Heng Yin[2]**
**Hao Su[1]**     **Xiaolong Wang[1]**

UC San Diego[1]     HKUST[2]

**Abstract:** We propose a sim-to-real framework for dexterous manipulation which can generalize to new objects of the same category in the real world. The key of our framework is to train the manipulation policy with point cloud inputs and dexterous hands. We propose two new techniques to enable joint learning on multiple objects and sim-to-real generalization: (i) using imagined hand point clouds as augmented inputs; and (ii) designing novel contact-based rewards. We empirically evaluate our method using an Allegro Hand to grasp novel objects in both simulation and real world. To the best of our knowledge, this is the first policy learning-based framework that achieves such generalization results with dexterous hands. Our project page is available at `https://yzqin.github.io/dexpoint`.

**Keywords:** Dexterous Manipulation, Point Clouds, Sim-to-Real

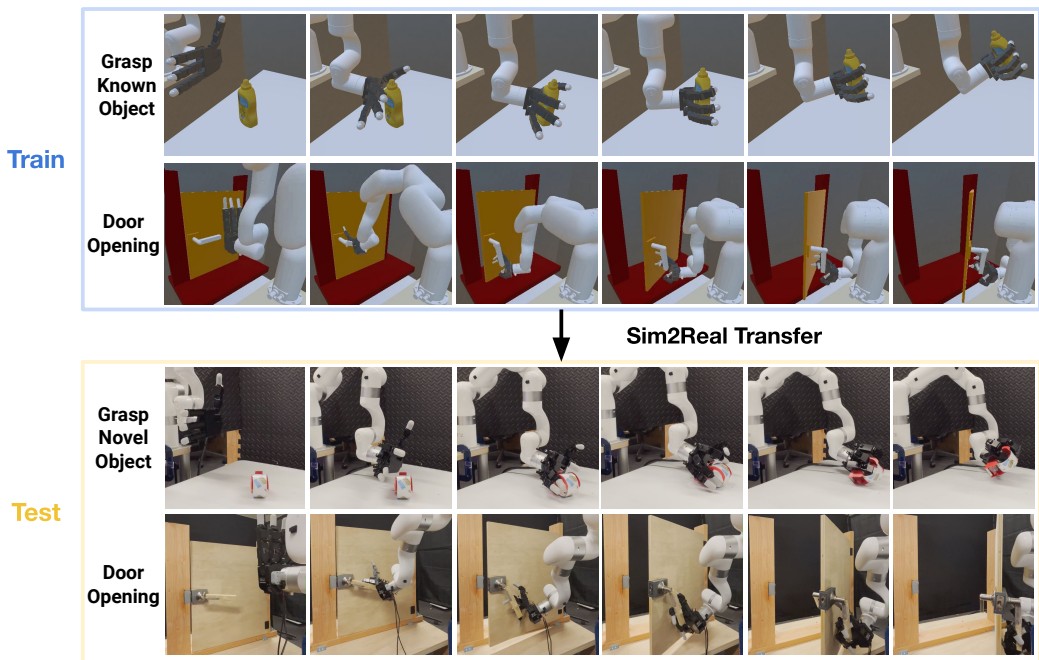

Figure 1: We introduce a reinforcement learning method which takes the point cloud as input for two manipulation tasks: grasping and door opening. By introducing several techniques in the policy learning process, our point cloud based policy trained purely in simulation can successfully generalize to novel objects and transfer to real-world without any real-world data.

---

* indicates equal contribution.

6th Conference on Robot Learning (CoRL 2022), Auckland, New Zealand.

# 1 Introduction

Dexterous manipulation has remained to be one of the most challenging problems in robotics [1]. While multi-finger hands create ample opportunities for robots to flexibly manipulate objects in our daily life, the nature of the high degree of freedom and high-dimensional action space creates significant optimization challenges for both search-based planning algorithms and policy learning algorithms. Recent efforts using model-free Reinforcement Learning have achieved encouraging results on complex manipulation tasks [1, 2]. However, it still faces many challenges in generalizing to diverse objects and being deployed on multi-finger hand in the real-world.

For example, the dexterous manipulation framework proposed by OpenAI et al. [1] can solve in-hand manipulation of Rubik's Cube with RL and transfer to the real robot hand. However, the policy is only trained with one particular object and it is not able to *generalize to diverse objects*. To achieve cross-object generalization, recent efforts proposed to learn robust 3D point cloud representations [3, 4, 5, 6] with diverse objects using RL in simulation. While point cloud input has also been shown easier for Sim2Real transfer [7] given its focus on geometry instead of texture, the assumption on the access of complete object point clouds and ground truth states limit the transferability of above methods to the *real robot deployment*. Among these works, Chen et al. [3] showed that cross-object generalization is achievable in simulation without knowing the shape of the object to grasp, but its requirement of real-time access to the object states is itself a very challenging robot perception problem, especially under large occlusions during hand object interaction.

In this paper, we provide a sim-to-real reinforcement learning framework for generalizable dexterous manipulation, using two tasks with the Allegro Hand[8]: (i) object grasping where the test objects has not been seen during training; (ii) door opening where the test doors have levers of novel shape that has not been used in training. The tasks are visualized in Figure 1.

the robot needs to first rotate the lever to unlock the door latch and then pull the lever in a circular motion to open the door.

We perform our studies by training a point cloud based reinforcement learning policy in the grasping and door opening task. With this approach, we list the three key discoveries of our framework for learning generalizable point cloud policy below:

(i) We justified that it is possible to achieve *direct sim-to-real transfer* for a dexterous manipulation policy with category-level generalizability when we use point cloud as the data representation.

(ii) Raw point clouds captured by sensors often come with heavy occlusions and noise: only a very small portion of the points from the observation are representing the robot fingers. We propose to *imagine* the complete robot finger point clouds according to the robot kinematic model and use them to augment the occluded real point cloud observations. We find that explicitly augmenting the input by *imagined* points can help achieve better robustness and sample efficiency for reinforcement learning.

(iii) Different from existing works that add contact information to the input of RL, we design a novel reward using contact pair information without adding contact to the observation. This practice remarkably improves sample efficiency as well as learning stability and avoids the dependency on contact sensor that is often unavailable for real robot models.

# 2 Related Work

**Dexterous Manipulation.** Dexterous manipulation aims to enable robotic hands to achieve human-level dexterity for grasping and manipulating objects. Analytical methods have been proposed to adopt planning to solve this problem [9, 10, 11, 12, 13, 14, 15]. However, they rely on detailed object models, which are not accessible when testing on unseen objects. To mitigate this issue, researchers propose a learning-based method for dexterous manipulation [16]. Some methods [17, 1, 2, 3, 5] take a reinforcement learning (RL) approach, while another line of work [18, 19, 20, 21] also proposes to learn from demonstration using imitation learning (IL) to acquire the control policy. Recently, the combination of RL and IL [22, 23, 24, 25, 26, 4] has also shown encouraging dexterous manipulation results in simulation. However, most methods are either learning policy for one single object or assume access to object states produced by perfect detector which increases challenges to Sim2Real

transfer. In this paper, we surpass these limitations by introducing training on multiple objects and applying point cloud inputs for control.

**Point Cloud in Robotic Manipulation.** Point cloud representation has been widely applied in the robotics community. Researchers have studied matching the observed point clouds to an object in a grasping dataset [27] and executing the corresponding grasping action [28, 14]. For learning-based approaches, one line of research has focused on first estimating grasp proposals or affordance given the point cloud input and then planning accordingly for manipulation [29, 30, 31, 32, 33, 34, 35, 36]. While these methods are designed for parallel-jaw grippers, recent advancements have also been made for grasping with dexterous hands with similar approaches [37, 38, 39, 40]. However, this line of methods requires feedback from motion planning to estimate whether a grasp is plausible. Oftentimes, a stable grasp pose is proposed but it is not achievable by planning. To achieve efficient and flexible manipulation, a Reinforcement Learning policy with point cloud inputs is proposed [3, 5, 4], which allows the robot hand to flexibly adjust its pose while interacting with the object. However, these approaches still face challenges in transferring to the real robot, given the noisy, occluded point clouds in the real world, and training RL policy with noisy point clouds increases the optimization difficulty. In this paper, we propose to use imagined point clouds, contact information, and multi-object training to combat these problems and achieve Sim2Real generalization.

**Using Contact Information for Manipulation.** Humans can manipulate objects purely from tactile sensing without seeing them. This biological fact inspires researchers to integrate contact and tactile information into the learning pipeline [41, 42, 43, 44, 45, 46, 47]. For example, both tactile and visual information inputs are combined together in [46] for decision making. The tactile information is also utilized as inputs for RL-based manipulation policies [48, 49, 50, 23, 51]. For example, visual-tactile sensor is used with an Allegro hand in simulator for playing piano [51]. Instead of using contacts as inputs, some methods also use contact and tactile information as reward to encourage exploration and boost policy learning [49, 23]. Motivated by these works, we provide a novel design of reward based on each contact link of the robot hand, which encourages more reasonable grasping behavior. Our design does not require a real tactile sensor when training in simulation or deploying in the real robot.

## 3 Approach

Our objective is to train a generalizable point cloud policy on a dexterous robot hand-arm system that is able to grasp a wide range of objects or open an closed door with RL. We aims at Sim-to-Real transfer without any real-world training or data. During testing, the robot can only access the single-viewed point cloud and the robot proprioception data. As is discussed before, training such a policy comes with numerous technical challenges, including reward design and imperfect point cloud information. In this work, we propose a novel reward design technique based on contact and imagined point cloud model to deal with these challenges.

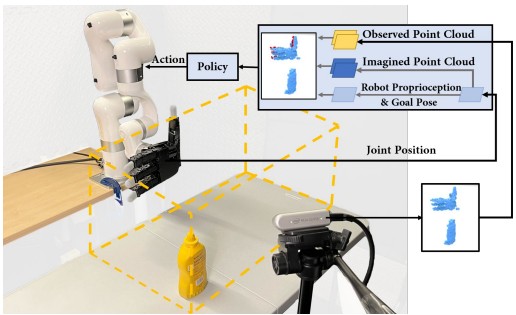

Figure 2: **Real-experiment Setup:** we use an Allegro Hand attached on an XArm6 and a RealSense D435 camera facing forward the robot.

**Preliminaries:** We model the dexterous manipulation problem as a Partially Observable Markov Decision Process (POMDP) $\mathcal{M} = (\mathcal{O}, \mathcal{S}, \mathcal{A}, \mathcal{R}, \mathcal{T}, \mathcal{U})$. Here, $\mathcal{O}$ is the observation space, $\mathcal{S}$ is the underlying state space, $\mathcal{A}$ is the action space, $\mathcal{R}$ is the reward function, $\mathcal{T}$ is the transition dynamics, and $\mathcal{U}$ generates agent's observation. At timestep $t$, the environment is at the state $s_t \in \mathcal{S}$. The agent observes $o_t \sim \mathcal{U}(\cdot|s_t) \in \mathcal{O}$. The agent takes action $a_t$ and receives reward $r_t = \mathcal{R}(s_t, a_t)$. The environment state at timestep $t + 1$ then transit to $s_{t+1} \sim \mathcal{T}(s_t, a_t)$. The objective of the agent is to maximize the return $\sum_{t=0}^{T} \gamma^t r_t$, where $\gamma$ is a discount factor.

**System Setup:** Many previous works on learning-based dexterous manipulation attach the hand to a fixed platform to simplify the experiment environment in the real world. In this work, we create a more flexible and powerful dexterous manipulation system which includes both the robotic hand and the arm (Figure 2). Concretely, we attach the Allegro Hand to an XArm6 robot. Allegro Hand

is a 16-DoF anthropomorphic hand with four fingers and XArm6 is a 6-DoF robot arm. We place a RealSense D435 camera at the right front of the robot to capture the point cloud. This setup brings additional challenges to RL exploration and Sim2Real deployment. We use SAPIEN [52] platform with uses a full-physics simulator to build the environment of the whole system. The simulation time step is $0.005s$ to ensure stable contact simulation. Each control step lasts for $0.05s$.

**Tasks and Objects:** In this paper, we expect our robot to perform the grasping task over a diverse set of objects and to open a locked door by rotating the lever. In the grasping task, we first select a random object from an object dataset and place it on the table. The robot is then required to move it to a target pose. Moreover, the robot should be able to generalize to different initial states, so we randomize both the initial pose and the goal pose for each trial. In simulation experiments, we use bottles and cans from both ShapeNet [53] dataset and YCB [54] dataset. In real-world experiments, we use novel unseen objects to test the policy. In the door opening task, the robot hand is required first to rotate the lever to unlock the door latch and then pull the lever in a circular motion. We use three doors to test the policies both in the simulation and the real world, only one is used for training and the other two are unseen doors. We also randomize the initial pose for each trial.

**Observation Space:** The observation contains both visual and proprioceptive information with four modalities: (1) Observed point cloud provided by the camera; (2) Proprioception signals of the robot including joint positions and end-effector position; (3) Imagined hand point cloud proposed in Sec. 3.2; (4) Object goal position provided in each trial. All the information is accessible on the real robot. The dimension of each observation modality is shown in Figure 3.

**Action Space:** The action is responsible for controlling both the 6-DOF robot arm and the 16-DOF hand. It has $6 + 16 = 22$ dimension in total. The robot arm is parametrized by the 6D translation and rotation of the end-effector relative to a reference pose. We use the damped least square inverse kinematics solver with a damping constant $\lambda = 0.05$ to compute the joint motion. Each finger joint of the Allegro hand is controlled by a position controller. Both robot arm and hand are controlled by PD controllers.

**Network Architecture:** The network architecture is visualized in Figure 3,

## 3.1 Reward Design with Oracle Contact

Since we aim to solve the dexterous manipulation problem with pure RL, the reward design is central to the method. We need a good reward function to ensure proper interaction between the robotic hand and the object. The whole interaction process consists of two phases. The first phase is to simply reach the object. The second phase is to grasp the object and move it to the target, which is more challenging. For the first phase, we encourage reaching with the following reaching reward:

$$r_{\text{reach}} = \sum_{\text{finger}} \frac{1}{\epsilon_r + d(\mathbf{x}_{\text{finger}}, \mathbf{x}_{\text{obj}})}. \tag{1}$$

Here, $\mathbf{x}_{\text{finger}}$ and $\mathbf{x}_{\text{obj}}$ are the Cartesian position of each fingertip and the target object. Note that $\mathbf{x}_{\text{obj}}$ is available when we perform training in simulation. However, using this reward alone cannot ensure proper contact for grasping. For example, the robot can touch the object with the back of the hand rather than the palm and then get stuck in this local minimum. Therefore, we introduce a novel contact reward to guarantee meaningful contact behavior:

$$r_{\text{contact}} = \textbf{IsContact}(\text{thumb}, \text{object}) \textbf{ AND } \left( \sum_{\text{finger}} \textbf{IsContact}(\text{finger}, \text{object}) \geq 2 \right). \tag{2}$$

This contact reward function outputs a boolean value in $\{0, 1\}$. It outputs $1$ only if the thumb is in contact with the object and there are more than one finger in contact with the object. Intuitively, it encourages the robot to cage the object within fingers. In this case, the robot can quickly find out stable grasping and lift the object to the target location. The lifting behavior is encouraged by

$$r_{\text{lift}} = r_{\text{contact}} \textbf{Lift}(\mathbf{x}_{\text{obj}}, \mathbf{x}_{\text{target}}). \tag{3}$$

The **Lift** function is basically in the form of Equation 1 and the main difference is that it will return a large reward value upon task completion. The overall reward function is a weighted combination of the terms above plus a control penalty:

$$\mathcal{R} = w_{\text{reach}} r_{\text{reach}} + w_{\text{contact}} r_{\text{contact}} + w_{\text{lift}} r_{\text{lift}} + w_{\text{penalty}} r_{\text{penalty}}. \tag{4}$$

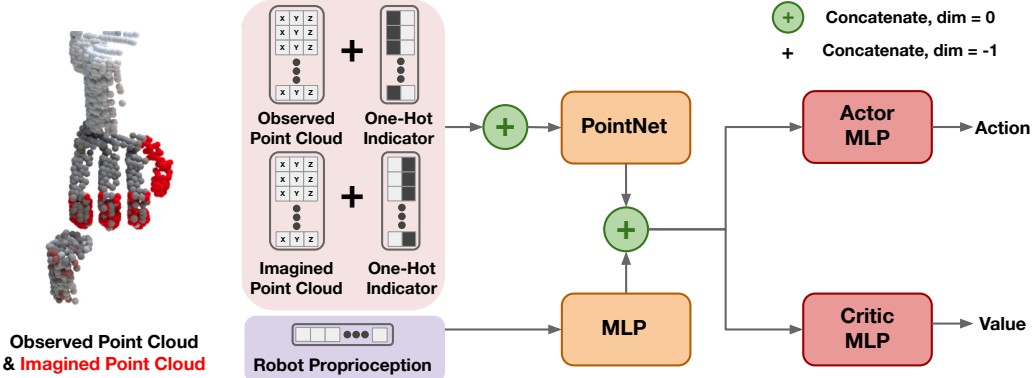

Figure 3: **Architecture:** our feature extractor takes the observed point cloud, imagined point cloud, robot proprioception, and goal pose as input to output a feature embedding. Both actor and critic take the same feature to predict action and value. The red point represented the imaged point cloud of robot hand. Note that our network does not require RGB information.

## 3.2 Imagined Hand Point Cloud

The usage of point cloud comes with two challenges. The first challenge is occlusion, which may occur to both the object under manipulation and the hand itself. When the robot hand is interacting with an object, the fingers may be occluded by the object. Since we do not assume tactile sensors in this work, this occlusion problem can be serious. The second challenge is the low point cloud resolution during RL training, where we can only use a limited number of points due to the memory limit. In this case, the number of points from hand finger may not be adequate to precisely capture the spatial relationship between the robot and the object. We propose a simple yet effective method to handle both issues in a unified manner. Our idea is to use an imagined hand point cloud in the observation to help the robot to *see the interaction*.

We provide one example in Figure 3. Black points indicate the point cloud captured by the camera, in which some important details of fingers are missing. These missing details provide crucial information of the interaction. Though such interaction information can also be inferred by combining the information from both the proprioception and visual input, we find that the best way is to synthesize these missing details. Concretely, we can compute the pose for each finger link via forward kinematics given the joint position from the robot joint encoder and the robot kinematics model. Then, we synthesize the imagined point cloud (blue points in Figure 3) by sampling the points from the mesh of each finger link. This process is possible in both simulation and the real world.

## 3.3 Training

We adopt Proximal Policy Optimization (PPO) [55] to train the agent in simulation, and then deployed to real without real-world fine-tuning. The network architecture is illustrated in Figure 3. Both value and policy networks share the same visual feature extraction backbone. We concatenate the observed point cloud with the imagined hand point cloud together as the input to the feature extractor. We also attach a one-hot encoding to each point which indicates whether it is observed or an imagined point.

## 4 Experiments

### 4.1 Experimental Setup

**Point Cloud Pre-processing:** To enable smooth transfer from simulation to real-world, we apply the same data preprocessing procedure to the point cloud captured by the camera. It involves four steps: (i) Crop the point cloud to the work region with a manually-defined bounding-box; (ii) Downsample the point cloud uniformly to 512 points; (iii) Add a distance-dependent Gaussian noise to the simulated point cloud to improve the sim2real robustness; (iv) Transform point cloud from the camera frame to the robot base frame using camera pose. In simulation, we use the ground-truth

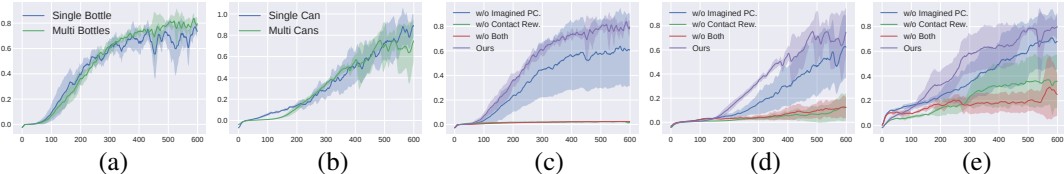

Figure 4: **Training Curves.** The left two plots show the single-object and multi-object training curve of (a) bottle category and (b) can category. The right three plots show the ablation results on the (c) grasping bottle (d) grasping can and (e) door opening. The x-axis is the training iterations and y-axis is the normalized episodic return. The shaded area indicates standard error and the performance is evaluated on five random seeds.

| Settings | Bottle | | Can | |
|---|---|---|---|---|
| | Known Obj. | Novel Obj. | Known Obj. | Novel Obj. |
| Single Obj. Training | $0.81 \pm 0.09$ | $0.60 \pm 0.06$ | $0.96 \pm 0.04$ | $0.63 \pm 0.18$ |
| Multi Obj. Training | $\mathbf{0.83 \pm 0.16}$ | $\mathbf{0.81 \pm 0.15}$ | $\mathbf{0.93 \pm 0.07}$ | $\mathbf{0.68 \pm 0.09}$ |

Table 1: **Experiment on Multi-object Training.** We evaluate the policy trained with single and multiple objects on bottle (upper two rows) and can (bottom two rows) categories with point cloud input. We test them on both known or novel objects. The success rate are reported on 5 seeds.

camera pose with multiplicative noise for frame transformation. In the real-world, we perform hand-eye calibration to get the camera extrinsic parameters.

**Evaluation Criterion:** We evaluate the performance of a policy by its success rate. For grasping tasks, a task is considered a success if $d_{ot} < 0.05m$ in simulation, where $d$ is the distance between object position and goal position. In real world, the task is considered as success if the XY position of the object is within 5cm from the target position and the height of the object is at least 15cm from table top. For the door opening, the task is considered as success if the door is opened to at least around 45 degrees.

**EigenGrasp Baseline:** We choose the EigenGrasp [56] as the grasp representation. Given an object mesh model, we use the GraspIt [57] to search valid grasp for Allegro Hand. Then, we use the RRTConnect [58] motion planner implemented in OMPL [59] to plan a joint trajectory to the pre-grasp pose and then plan a screw motion from pre-grasp pose to the grasp pose. Finally, we close all fingers based on searched grasp pose and lift the object to the target. Note that different from our approach, the baseline method **requires complete object model to search for grasps and ground-truth object pose** to align the grasp pose in the robot frame. To evaluate the performance of baseline on novel objects, we first build a grasp database on ShapeNet bottle and can categories using GraspIt. Given the sensory data of a new object, we search for the most similar objects in the dataset and use the query grasps for the novel object. Here we compare the performance of our method with baselines in the real-world.

**Training:** We train RL in two settings for grasping: (i) training on a single object (ii) training on multiple objects jointly. For the single object grasping, we perform experiments on both "tomato soup can" and "mustard bottle" from YCB. For the multi-object training, we choose 10 objects from the can or bottle categories of ShapetNet. We randomly choose one object to train for each episode in the multi-object training. Here, we use can and bottle as experimental subjects since they represent two different basic grasping patterns [60] for anthropomorphic hand: precision grasp and power grasp. For door opening, we only train the policy on the door with fixed lever geometry.

## 4.2 Comparison of Single-object and Multi-object Training

We plot the training curve of RL in Figure 4 (a) and (b). In general, our method can learn to grasp and move the object to the goal pose within 600 iterations, where each iteration contains 20K environment steps. Then we evaluate the policy trained on both known and novel objects, and the results are shown in Table 1. We run 100 trials to compute the average success rate. On grasping known objects, we find that agent trained on single-object outperforms the agent trained on multi-object by a small margin, for both learning efficiency and final success rate. However, agent trained on multi-objects does much better at grasping novel objects. Our results suggest that using multiple object during training is important, and is of great importance for novel object generalization.

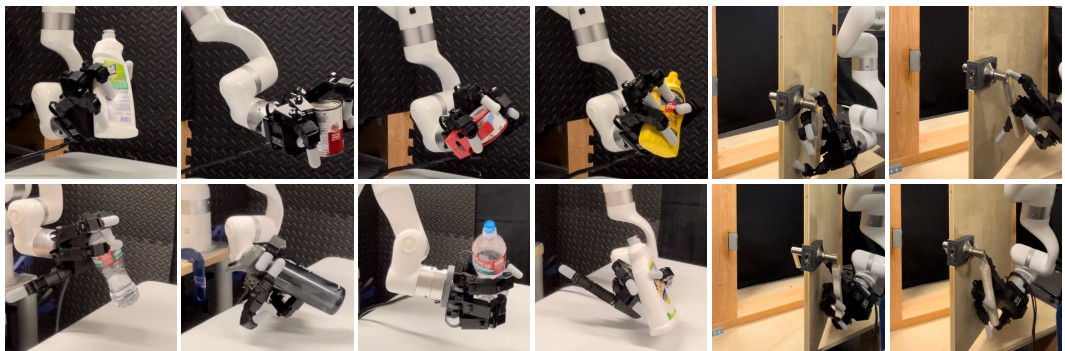

Figure 5: **Real-experiment:** We evaluate our point cloud policy on various unseen objects.

| Settings | Bottle | | Can | | Door | |
|---|---|---|---|---|---|---|
| | Known Obj. | Novel Obj. | Known Obj. | Novel Obj. | Known Obj. | Novel Obj |
| w/o Imagined PC. | $0.60 \pm 0.46$ | $0.56 \pm 0.51$ | $0.91 \pm 0.17$ | $0.63 \pm 0.07$ | $0.14 \pm 0.28$ | $0.11 \pm 0.27$ |
| w/o Contact Rew. | $0.00 \pm 0.00$ | $0.00 \pm 0.00$ | $0.06 \pm 0.08$ | $0.03 \pm 0.06$ | $0.21 \pm 0.26$ | $0.20 \pm 0.25$ |
| w/o Both | $0.00 \pm 0.00$ | $0.00 \pm 0.00$ | $0.00 \pm 0.00$ | $0.00 \pm 0.00$ | $0.00 \pm 0.00$ | $0.00 \pm 0.00$ |
| Ours | $\mathbf{0.83 \pm 0.16}$ | $\mathbf{0.81 \pm 0.15}$ | $\mathbf{0.93 \pm 0.07}$ | $\mathbf{0.68 \pm 0.09}$ | $\mathbf{0.92 \pm 0.06}$ | $\mathbf{0.79 \pm 0.11}$ |

Table 2: **Ablation Study:** we investigate the influence of contact-based reward design and imaged point cloud. We evaluate the success rate on both known and novel objects under four settings: (i) without imaged point cloud; (ii) without contact reward; (iii) without both; (iv) with both.

### 4.3 Ablation Results in Simulation

We ablate two key innovations of the work: the reward design with oracle contact and the imaged hand point cloud. We perform experiments on four different variants: (i) without imagined point cloud; (ii) without contact-based reward design; (iii) without both imagined point cloud and contact based reward design; (iv) our standard approach with both techniques. Note that the variant (iii) is an approximation of [3] in our environments and tasks. We compare both the learning curve and the evaluation success rate of these four variants. For grasping task, Figure 4 (c) and (d) show the results on bottle and can categories, and Table 2 shows the success rate. Our findings can be summarized as follows.

First, we find that contact reward information is of vital importance for training the point cloud RL policy on the multi-finger robot hand. Without using contact reward, the agent can hardly learn anything (red and green curve in the figure) and get nearly zero success rate during evaluation for both bottle and can categories. By encouraging the contact between fingers and object, the RL agent can avoid getting stuck in local minimums and learn meaningful manipulation behavior.

Second, the imagined point cloud can also improve the training and test performance for both categories, though it is not so important as the contact reward. As is shown in the learning curves of Figure 4 (c) and (d), the policy utilizing the imagined point cloud as input can learn faster in the early stage of the training and show much smaller variances. An interesting fact is that imagined point cloud is more beneficial for the bottle category than the can category. One possible reason is that grasping a bottle requires multi-finger coordination to perform a power grasp. Such coordination is highly dependent on the detailed finger information provided by the imagined point cloud. The imagined point cloud can help the agent to better see the fingers even if they are occluded by the object, e.g., fingers behind the object.

The experiments on the door opening task also support these findings. As shown in Figure 4 (e), the policy trained with contact based reward and imaged hand point cloud outperform other ablation methods, which also demonstrate the effectiveness of our two key design. Compared with the grasping experiments, we can observe larger variations during policy training. One possible reason is that door opening suffers from heavier occlusion than grasping, when the door lever is grasped by the robot hand, which influence the temporal consistence of PointNet feature.

### 4.4 Real-World Evaluation

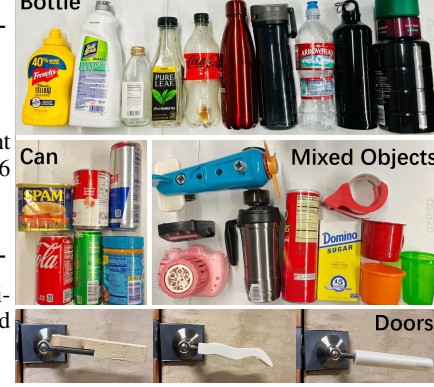

| Method | Bottle | Can | Mixed Category |
|---|---|---|---|
| EigenGrasp Oracle | 0.66 | 0.45 | *N/A* |
| EigenGrasp | 0.50 | 0.41 | *N/A* |
| Single Obj. Train | $0.75 \pm 0.06$ | $0.75 \pm 0.08$ | *N/A* |
| Multi Obj. Train | $\mathbf{0.87 \pm 0.03}$ | $\mathbf{0.83 \pm 0.13}$ | $\mathbf{0.73 \pm 0.12}$ |

Table 3: **Real-World Grasping Experiment**: This experiment consist of 3 categories, which incorporate 26 objects: 10 bottles, 6 cans, and 10 other objects in multiple mixed categories.

| Settings | Original Door | Novel Door 1 | Novel Door 2 |
|---|---|---|---|
| Single Door Train | $0.72 \pm 0.07$ | $0.60 \pm 0.03$ | $0.67 \pm 0.01$ |

Table 4: **Real-World Door Opening Experiment**: This experiment consist of 3 different doors, the first one on the left is used for training and the other two are for testing.

We perform sim2real experiments to evaluate the performance of our method in the real world. As is shown in Figure 2, we attached an Allegro hand onto a XArm-6 robot arm to grasp the object on the front table. We apply the same data-preprocessing steps for both simulated environment and real-world as mentioned in Sec.4.1.

The task execution sequence is visualized on the bottom row of Figure 1. Both Single Obj. Training and Multi Obj. Training will be evaluated in the corresponding category, while policies training with multiple objects even evaluated in the Mixed category with unseen objects. We run 10 independent trials seeds for each object-policy pair.

The real-world evaluation results are shown in Table 3 and Table 4. The policy trained in the simulator with point cloud input can directly transfer to the real-world without fine-tuning. For both two tasks, our policy can even deploy on objects that have never been seen during the training. Moreover, We find that for grasping, training on multiple objects can ensure better performance than single-object training. Compared with the EigenGrasp baseline shown in Table 3, our policies trained on both single-object and multi-object performs better, even if the EigenGrasp baseline use the object model information. Since EigenGrasp + motion planning is a open-loop manipulation policy, small error in object and scene modeling, e.g. initial object pose or object geometry, can lead to a failure in final grasp results. In contrast, our methods works in a closed-loop fashion with point cloud observation, which does not require privilege knowledge about the object.

## 5 Conclusion and Limitation

**Limitations** In our experiments, we only train and test our method on two tasks, which limits the scope of the proposed method. More dexterous manipulation tasks will be our future research direction. Another potential improvement of our method is to use Recurrent Neural Network and temporal information for policy networks. It will enable us to do long-horizon tasks.

**Conclusion** To the best of our knowledge, our approach is the first work to train a dexterous manipulation reinforcement learning policy with point cloud inputs that can transfer to the real world. We justified that direct sim-to-real transfer is possible for two manipulation tasks with point cloud representation.

### Acknowledgments

This work was supported, in part, by grants from NSF CCF-2112665 (TILOS), NSF 1730158 CI-New: Cognitive Hardware and Software Ecosystem Community Infrastructure (CHASE-CI), NSF ACI-1541349 CC*DNI Pacific Research Platform, the Industrial Technology Innovation Program (20018112, Development of autonomous manipulation and gripping technology using imitation learning based on visualtactile sensing) funded by the Ministry of Trade, Industry and Energy of the Republic of Korea, and gifts from Meta, Google, Qualcomm. We also thanks Fanbo Xiang for suggestion on shader setting in SAPIEN, Ruihan Yang for helpful discussion on RL training.

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
