# OpenReview forum: "DexPoint: Generalizable Point Cloud Reinforcement Learning for Sim-to-Real Dexterous Manipulation"
_robot-learning.org/CoRL/2022/Conference — CoRL 2022 Poster_

### Official Review · Reviewer_6SGW · 2022-07-13

**Originality:** Good
**Technical Quality:** Good
**Clarity Of Presentation:** Very Good
**Impact:** 3

**Recommendation:**

Weak Accept: I recommend accepting the paper, but will not argue for my recommendation if the majority of other reviewers have a different opinion.

**Summary:**

The paper proposes a method for robotic grasping with a dexterous robot hand in the real world. The method is point-cloud based and fully trained in simulation. The authors propose a shaped reward, including contact information between robotic hand and the object to grasp. Furthermore they propose to construct the point-cloud of the robotic hand from proprioceptive information and use this imagined point cloud as an additional input to their method.
The authors ablate their modelling choices in simulation and  demonstrate that their trained method can be transferred to the real world without fine-tuning.

**Issues:**

Formulation of loss:
- line 140:  how is $x_{obj}$ is defined? Is it the center of the object?
- eq.2: How is inContact defined? How is it determined if contact was established?

Experiments:
- There is no comparison to a baseline.
- The evaluation is performed only on a very limited set of objects (bottles and cans) and only on four objects of type bottle.
- Table 2: The caption seems to be wrong. There is no learning curve visible here.


**Quality Of The Limitations Section:**

Additional details required

**Reviewer Expertise:**

4: The reviewer is confident but not absolutely certain that the evaluation is correct

**Robotics Focus:**

Sufficient demonstration on hardware

**Strengths And Weaknesses:**

Strengths:
- The authors demonstrate that their method can be transferred to the real world without fine-tuning.
- The formulation of the reward seems to be useful and lead to good results.
- The idea of adding the imagined gripper point cloud seems improve the training.

Weaknesses:
- The evaluation the authors perform is rather limited: They only train and evaluate on cans and bottles in simulation. In the real world the evaluation is even more limited, as the authors only evaluate on 4 different bottles.
- There is no comparison to other methods like e.g. Unigrasp [L. Shao et al., "UniGrasp: Learning a Unified Model to Grasp With Multifingered Robotic Hands," in IEEE Robotics and Automation Letters, vol. 5, no. 2, pp. 2286-2293, April 2020, doi: 10.1109/LRA.2020.2969946].

**Summary Of Recommendation:**

The ideas seem nice and promising, but the evaluation is not very thorough. There is no comparison to any baseline (e.g. Unigrasp). Also the evaluation is performed only on very view objects of limited variability. In particular the real world experiments are performed only four objects of type bottle.

---

> ### Author Response · Authors · 2022-08-27
> **Response to Reviewer 6SGW**
>
> **Q**: The evaluation the authors perform is rather limited: They only train and evaluate on cans and bottles in simulation. In the real world the evaluation is even more limited, as the authors only evaluate on 4 different bottles.
>
> **A**: We perform new real-robot experiments to increase object diversity. We not only include in-domain objects in the same category but also **out of training domain** objects in different categories. The new real-world experiments consist of 3 categories, which incorporate 26 objects: **10 bottles, 6 cans, and 10 other objects mixed with multiple categories**, shown in [this anonymous link](https://drive.google.com/uc?id=1JI3XXIkTMqjbTg5exqni7E-sxpuCq-32).  These objects require different grasp patterns with diverse shapes. We believe that the new experiments better show the generalizability of our method to novel objects.
>
> The updated version of Table 3 in the original submission is as follows:
>
> |Test Category |Single Obj. Training|Multi Obj. Training|
> |----------|--------|------|
> | Bottle Category | $0.75 \pm 0.06$ | $\mathbf{0.87 \pm 0.03}$ |
> | Can Category | $0.75 \pm 0,08$ | $\mathbf{0.83 \pm 0.13}$ |
> | Mixed Category | N/A | $\mathbf{0.73 \pm 0.12}$ |
>
> **Details on the mixed category set**:
> The mixed categories are outside training categories such as mug, toys, camera, box, and tap. We train policies and both bottle and can categories simultaneously with different random seeds and evaluate these policies on the mixed category. Our policy achieves a reasonable success rate even tested on the unseen out-of-distribution mixed category objects.
>
> We have also provided the videos of the policy deployment on these objects on our  [project page](https://dexpc.github.io/).
>
> **Q**: There is no comparison to other methods like e.g. Unigrasp [L. Shao et al., "UniGrasp: Learning a Unified Model to Grasp With Multifingered Robotic Hands," in IEEE Robotics and Automation Letters, vol. 5, no. 2, pp. 2286-2293, April 2020, doi: 10.1109/LRA.2020.2969946].
>
> **A**: Thanks for your suggestion. We agree it is essential to compare with previous methods. Here we choose to use the grasp generation method, which is also used in UniGrasp to generate the ground-truth training data based on contact modeling. In the following text, we will first describe the baseline method and then show experimental results and analysis with the baseline approach.
>
> We choose the widely-used EigenGrasp [1] as the grasp representation. Given an object mesh model, we use the well-known GraspIt [2] to search valid grasp for AllegroHand. Then, we use the RRTConnect [3] motion planner implemented in OMPL [4] to plan a joint trajectory to the pre-grasp pose and use a screw motion controller to move the robot from pre-grasp pose to the grasp pose. Finally, we close all fingers in a direction specially designed for Allegro Hand and lift the object up to the target. Note that different from our approach, the baseline method requires **complete object model** to search for grasps and **ground-truth object pose** to align the grasp pose in the robot space. So we call this grasping baseline an oracle. To evaluate the performance of baseline on novel objects, we first build a grasp database on a ShapeNet bottle and can categories using GraspIt. Given the sensory data of a new object, we search for the most similar objects in the dataset and use the query grasps for the novel object. Here we compare the performance of our method with baselines in the real world.
>
> The success rate comparison is listed in the following table. Note that we include more objects for testing. The bottle category includes 10 bottles and the can category includes 6 cans for testing as shown in [this anonymous link](https://drive.google.com/uc?id=1JI3XXIkTMqjbTg5exqni7E-sxpuCq-32).
>
> |Method |Bottle Category|Can Category|
> |----------|--------|------|
> | EigenGrasp (oracle) | $0.66$ | $0.50$ |
> | EigenGrasp | $0.45$ | $0.41$ |
> | Ours | $\mathbf{0.87}$ | $\mathbf{0.83}$ |
>
> [1] Dimensionality reduction for hand-independent dexterous robotic grasping
>
> [2] GraspIt!: A Versatile Simulator for Grasp Analysis
>
> [3] RRT-Connect: An Efficient Approach to Single-Query Path Planning
>
> [4] The Open Motion Planning Library
>
> **Q**: line 140: how is $X_{obj}$ defined? Is it the center of the object?
>
> **A**: It is defined as the center of an object bounding box.
>
> **Q**: eq.2: How is inContact defined? How is it determined if contact was established?
>
> **A**: It is defined by checking the contact force between a pair of links. If the contact force is larger than a given threshold, i.e. 2N, then two links are considered in contact.
>
> **Q**: Table 2: The caption seems to be wrong. There is no learning curve visible here.
>
> **A**: Thanks for pointing this out. We will update the caption of Figure 2.

---

### Official Review · Reviewer_aTKU · 2022-07-25

**Originality:** Fair
**Technical Quality:** Good
**Clarity Of Presentation:** Good
**Impact:** 3

**Recommendation:**

Weak Accept: I recommend accepting the paper, but will not argue for my recommendation if the majority of other reviewers have a different opinion.

**Summary:**

The paper proposes two techniques to learn dexterous manipulation policies with point clouds as observation. The authors propose using Reinforcement Learning to learn the policy and evaluate the performance of the learned policy when transferred from simulation to reality.

The paper's main contributions are:
1. Show that it is possible to achieve sim-to-real transfer for dexterous grasping policies when using point cloud as observations.
2. Two techniques to guarantee proper learning:
         (a) a Pointcloud augmentation to deal with possible camera occlusions
         (b) a hand-object contact reward function.


**Issues:**

1. Adding a new experiment on a different task. It could be related to grasping, such as opening a drawer.
2. Evaluating the benefits of the proposed techniques with different RL algorithms. The method should work, but the paper would be more complete if an experiment with SAC and TD3 are added.

**Quality Of The Limitations Section:**

Additional details required

**Reviewer Expertise:**

4: The reviewer is confident but not absolutely certain that the evaluation is correct

**Robotics Focus:**

Sufficient demonstration on hardware

**Strengths And Weaknesses:**

In the reviewer's view, the strengths of the paper are:
1. The demonstration of the utility of point cloud observation for sim-to-real transfer.

In the reviewer's view, the weaknesses of the paper are:
1. The paper is very task specific. The authors present the method they have used to train a reinforcement learning agent to grasp objects given point clouds. To do so, they introduce a reward function and the observation augmentation methods but it is not clear if these methods would also benefit in other possible dexterous manipulation tasks as it has been only evaluated for grasping. Evaluating the method in additional manipulation tasks would be beneficial to show the benefit of the proposed method.
2. The solution is validated only with PPO. It would be interesting to observe the performance of the proposed method with other reinforcement learning algorithms.


**Summary Of Recommendation:**

The paper presents a problem and provides a solution to it. The authors show that the method works for different objects and show that the method transfer to real robots. The paper itself seems very specific to the problem of learning through reinforcement learning and a grasping policy through point clouds. But, it is properly executed and the authors show the performance of the proposed method and validate the benefit of the proposed techniques.

Therefore, I suggest a weak acceptance.

---

> ### Author Response · Authors · 2022-08-27
> **Response to Reviewer aTKU**
>
> **Q**: it is not clear if these methods would also benefit in other possible dexterous manipulation tasks as it has been only evaluated for grasping. Evaluating the method in additional manipulation tasks would be beneficial to show the benefit of the proposed method. Adding a new experiment on a different task. It could be related to grasping, such as opening a drawer.
>
> **A**: Thank you for the feedback. We indeed have focused mainly on grasping with a dexterous robot hand in the original submission. However, our approach provides a framework that can be generalized to other manipulation tasks. To validate this, we additionally provide a **Door Opening** task here. We have provided new experiment results for the door opening task and uploaded the **policy execution videos** to our [project page](https://dexpc.github.io/). This door opening task requires the robot hand to first rotate the lever to unlock the door latch and then pull the lever in a circular motion to open the door. This task requires delicate coordination between hand wrist motion and finger motion, thus we consider it a challenging dexterous manipulation task. Here we show the evaluation results of door opening tasks in both simulation and the real world. We randomized the position of the door relative to the robot base for each trail in both sim and real. The task is considered a success if the door is opened to at least 45 degrees. Here we show the success rate of our method on door opening tasks in both simulator and real world.
>
> | | Success Rate
> |----------|--------|
> | Simulator | $0.92 \pm 0.06$ |
> | Real World | $0.65 \pm 0.05$ |
>
>
>
> **Q**:The solution is validated only with PPO. It would be interesting to observe the performance of the proposed method with other reinforcement learning algorithms. Evaluating the benefits of the proposed techniques with different RL algorithms. The method should work, but the paper would be more complete if an experiment with SAC and TD3 are added.
>
> **A**: We have also tried TRPO as the RL algorithm for policy training. We observe that the overall performance of PPO and TRPO is very similar while PPO learns faster during the initial training. The comparison between TRPO and PPO and both bottle and can categories are shown in [this anonymous link](https://drive.google.com/uc?id=1zErrBC9jE2wzlvAFal7tiM11d1YM2Izw). As shown in this figure, TRPO shows smaller variance than PPO but learns little bit slower.
> As for off-policy learning algorithms, e.g. SAC and TD3, they are often not used when dexterous robot hand are involved in the task due to high dimensional action space. As discussed in [1], on-policy scale better to high dimensional spaces. So we do not choose off-policy approach as our learning algorithm.
>
> [1] Learning Complex Dexterous Manipulation with Deep Reinforcement Learning and Demonstrations

---

### Official Review · Reviewer_BaTP · 2022-07-31

**Originality:** Good
**Technical Quality:** Good
**Clarity Of Presentation:** Very Good
**Impact:** 3

**Recommendation:**

Weak Accept: I recommend accepting the paper, but will not argue for my recommendation if the majority of other reviewers have a different opinion.

**Summary:**

This work proposes a policy learning framework for utilizing multi-fingered hands, such as the Allegro platform, to grasp rigid objects. The input corresponds to a point cloud, which is augmented with proprioceptive and object pose information to construct the observation space. The action space consists of the arm’s 6 joint angles and the end-effector's 16 DoFs. Relatively to prior efforts, this work introduces a contact-based reward function (which is where most of the argued benefits arise from) and an “imagination”- based point cloud augmentation technique (for mitigating occlusions and sensor noise). The final outcome is category-based generalization to novel object instances and zero-shot sim2real transfer for grasping. An ablation study has been conducted (in simulation) to compare different training strategies and how training over multiple objects affects the approach. The performance in the real world was tested with four objects.

**Issues:**

Change the title of the paper to replace “Dexterous Manipulation” with “Multi-fingered Grasping”.

This algorithm is tested on a quite standard setup (single tabletop object, uncluttered, a commonly-used hand) and has virtually no inference comparison point. So, papers like the ‘Multi-fingered Active Grasp Learning’ by Lu et al., ‘Learning Continuous Grasping Function with a Dexterous Hand from Human Demonstrations’ by Ye et al. (admittedly it has been published after the submission deadline)), ‘Multifingered Grasping Based on Multimodal Reinforcement Learning’ by Liang et al.  and other similar works could be included in the related work section. One of them could potentially be used as a comparison point to this algorithm’s performance.

It would be appreciated if the authors provided failure examples and look into which part of their method is causing them and in which scenarios.

How many training iterations would be required if somebody would want to train for a new object category?

Please provide some justification for the following choices: a) The Object/Goal Pose Rotational components are not included in the reward term. b) Only two fingers are needed for the contact reward term to be active.

The authors should add legends to their plot diagrams (including measurement units).


**Quality Of The Limitations Section:**

Limitations are not well addressed

**Reviewer Expertise:**

4: The reviewer is confident but not absolutely certain that the evaluation is correct

**Robotics Focus:**

Sufficient demonstration on hardware

**Strengths And Weaknesses:**

Strengths:
- The problem that the paper attempts to solve involves a high-dimensional control challenge, due to the presence of a multi-fingered hand. For such end-effectors, which bring the promise of dexterous manipulation, grasping singulated objects (which is the focus of this work) can still be challenging especially as object variability increases.
- The ideas on the “oracle contact” and the imagined hand point cloud seem reasonable for the proposed objective.
- The authors have tested their approach in real-world robot experiments and seemed to have achieved zero-shot Sim2Real transfer, which is highly desirable.
- The paper is mostly clearly written, with proper formalism and an accompanying ablation in simulation. The visualizations are helpful.
- The related work section is rather comprehensive and covers the important subareas related to this work.

Weaknesses:
- The expression "Dexterous Manipulation" used in the title is rather misleading. It should be replaced by the term ‘Multi-fingered Grasping’, which is what this work is focusing on.
- The fact that the method is both trained and tested only on 2 distinct object categories + the nature of the contact-based reward may imply that the approach learns a shape prior and that this is the main reason this method succeeds. What happens in object categories significantly outside this shape prior still remains open for interpretation.
-The reward function seems to have multiple components that are aggregated together with a rather naive weighting scheme. The reward structure does not seem to reflect the structure of the task (e.g., first approach the object and then lift it) as the weighting scheme seems to be constant over the duration of the trajectory instead of time-dependent. It would be helpful to further explore how the different components can be integrated into an effective reward function and the role of the weights in this process.  Some of this discussion is left for the supplementary material but it would be better if the penalty term in the overall reward, as well as the multi-task weights’ scheduling scheme, were introduced in the main paper.
- In contrast to some related efforts (e.g., [3]), access to the object geometries seems to be a requirement for this work during training.
- The “Limitations” section is more of ‘Future Work’ one. It would be better if the authors acknowledged in a more comprehensive manner the limitations (e.g., there is still a gap until the grasping problem targeted is solved).
- The experiments in the ablation study between single and multiple objects do not communicate a clear message. In particular, the training curves for the two object categories provided do not give a clear indication which method is better and why. Since this is referring to simulation-based experiments, adding more categories may be helpful to evaluate a pattern.
- There is a mismatch between what is described for the Multi-object case between the main paper and the appendix. In the second one, only the ShapeNet objects are mentioned, while in the first one the authors mention that they randomly sample from both datasets.
- According to the ablation study, the approximation of the previous, related approach ([3]) achieves 0% success rate. How does this match the results presented in that work, which focused on in-hand object reorientation and must have included contacts.
- The provided videos do not visualize the goal pose at each test case (like in the paper’s diagram), which makes it harder for the reviewer to evaluate the task’s success/failure. Furthermore, there is no video that clearly showcases the effect of adding solely the contact-based reward term, the strongest point of the paper.


**Summary Of Recommendation:**

This a rather well-written paper, with solid fundamentals, aiming to solve a high-dimensional grasping challenge. Both contributions presented are reasonable and presented in sufficient detail. The paper provides real-world experiments and demonstrates Sim2Real transfer. The results are intuitive, even though there are some points that would need further clarification per the comments above. The quantitative results of the method are well-organized and indicate an improvement in terms of grasping success even though there are still failure cases.. Some additional discussion of the qualitative results would be helpful.

---

> ### Author Response · Authors · 2022-08-27
> **Response to Reviewer BaTP (Weaknesses Part)**
>
> Thank you for the helpful comments and valuable feedback! We address individual comments in the following.
>
> **Q**: The expression "Dexterous Manipulation" used in the title is rather misleading.
>
> **A**: Thank you for the feedback. We indeed have focused mainly on grasping with a dexterous robot hand in the original submission. However, our approach provides a framework that can be generalized to other manipulation tasks. To validate this, we additionally provide a **Door Opening** task here. We have provided new experiment results for the door opening task and uploaded the **policy execution videos** to our [project page](https://dexpc.github.io/). This door opening task requires the robot hand to first rotate the lever to unlock the door latch and then pull the lever in a circular motion to open the door. This task requires delicate coordination between hand wrist motion and finger motion, thus we consider it a challenging dexterous manipulation task.
>
>
> **Q**: The fact that the method is both trained and tested only on 2 distinct object categories + the nature of the contact-based reward may imply that the approach learns a shape prior and that this is the main reason this method succeeds.
>
> **A**: We perform new real-robot experiments to increase object diversity. We not only include in-domain objects in the same category but also **out of training domain** objects in different categories. The new real-world experiments consist of 3 categories, which incorporate 26 objects: **10 bottles, 6 cans, and 10 other objects mixed with multiple categories**, shown in [this anonymous link](https://drive.google.com/uc?id=1JI3XXIkTMqjbTg5exqni7E-sxpuCq-32).  These objects require different grasp patterns with diverse shapes. We believe that the new experiments better show the generalizability of our method to novel objects.
>
> **Q**: The reward structure does not seem to reflect the structure of the task.
>
> **A**: We would like to clarify that our grasping reward function includes three stages: approaching objects, building contact, and lifting up. The RL agent can only receive the reward next when the previous stage is solved. Our stage-wise reward design also encodes the task structure and encourages the agent to finish the task for each stage in the appropriate order. Here we provide the pseudo-code of the reward function:
>
> ```python
> reward = compute_reach_reward(robot_fingers, object) * w_reach
> is_contact = env.count_contact(robot_fingers, object) > 2 && env.count_contact(robot_thumb, object) > 1
>
> if is_contact:
>     reward += w_contact
>     lift = compute_height_difference(object, target) # Only provide lift reward if the finger-object is in contact, otherwise the robot hand will exert a large force to knok the object into air
>     reward += lift * w_lift
>
> reward += l2_norm(action) * w_penalty # Penalize large action
> ```
>
>
> **Q**: Comparison to [3]
>
> **A**: As mentioned in paper [3] in Sections 2.1 and 2.2.1, for the generalizable policy across different instances, their observation space incorporates **oracle object state information that is not directly accessible in the real-world**: object position and orientation of each step is provided in the observation even for their student policy. In fact, the orientation of objects can be defined only when the geometry is given. Thus [3] indeed utilize geometry information implicitly inside the state space. In our experiments, we do not include object pose inside observation. Thus, the performance of ablation on [3] achieves a very low success rate in the real world.
>
> **Q**: The “Limitations” section is more of ‘Future Work’ one.
>
> **A**: Thank you for the suggestion. We will rephrase the limitation section.
>
> **Q**: The experiments in the ablation study between single and multiple objects do not communicate a clear message.
>
> **A**: We want to emphasis that multi-object training is beneficial to generalization. Our new experiments shows that training on mixed object dataset can even provide policy the ability to generalize to novel categories.
>
> **Q**: There is a mismatch between what is described for the Multi-object case between the main paper and the appendix.
>
> A: Sorry for the confusion. The single object experiments in both sim and real are single can or mug from the YCB dataset.  For multi-object experiments in simulation, we only sample objects from the ShapeNet. For real-world experiments,  we sample objects from both YCB and other object sets. We will rewrite this part in the main paper to make it more clear.
>
> **Q**: The provided videos do not visualize the goal pose at each test case. Furthermore, there is no video that clearly showcases the effect of adding solely the contact-based reward term, the strongest point of the paper.
>
> **A**: Thanks for your information. The updated video in our [project page](https://dexpc.github.io/) now visualizes the target pose in the simulated environment.

---

> ### Author Response · Authors · 2022-08-27
> **Response to Reviewer BaTP (Issues Part)**
>
> **Q**: This algorithm is tested on a quite standard setup (single tabletop object, uncluttered, a commonly-used hand) and has virtually no inference comparison point. So, papers like the ‘Multi-fingered Active Grasp Learning’ by Lu et al., ‘Learning Continuous Grasping Function with a Dexterous Hand from Human Demonstrations’ by Ye et al. (admittedly it has been published after the submission deadline)), ‘Multifingered Grasping Based on Multimodal Reinforcement Learning’ by Liang et al. and other similar works could be included in the related work section. One of them could potentially be used as a comparison point to this algorithm’s performance.
>
> **A**: Thanks for your suggestion. We agree it is essential to compare with previous methods. In the updated version, we have included a grasp generation baseline using EigenGrasp with GraspIt, which is very similar to *Multi-fingered Active Grasp Learning*. The second paper *Learning Continuous Grasping Function with a Dexterous Hand from Human Demonstrations* focus more on a open-loop setting where the trajectory are planned ahead before execution. In comparison, we framework provides a close-loop policy to predict the next action re-actively.
>
> **Q**:It would be appreciated if the authors provided failure examples and look into which part of their method is causing them and in which scenarios.
>
> **A**: Thanks for your comment. We have updated the failure examples of two typical failure cases on our [project page](https://dexpc.github.io/).
>
> **Q**: How many training iterations would be required if somebody would want to train for a new object category?
>
> A: We find that 600 iterations are sufficient for most objects. If the policy can not get success after 600 iterations, the policy will be trapped in the local minimum and fail to solve the task even with more training iterations.
>
> **Q**: Please provide some justification for the following choices: a) The Object/Goal Pose Rotational components are not included in the reward term. b) Only two fingers are needed for the contact reward term to be active.
>
> **A**: `Why we do not consider rotation components in the reward term`: many objects in the can and bottle category are symmetrical or nearly-symmetrical (not exactly symmetrical but hard to tell considering the noise of depth cameras). If we want to include a rotation reward term, we need to annotate the symmetry type for each instance and compute the reward based on symmetry type, which limits our application to novel instances.
>
> `Why only two fingers are needed`: For a pinch grasp, it only needs two fingers while the power grasp uses all fingers to hold the object. We do not want to restrict the grasp pattern learned by the policy. If we require more contact, then the agent can not learn pinch grasp behavior.
>
>
> **Q**: The authors should add legends to their plot diagrams (including measurement units).
>
> **A**: We thank the reviewers’ suggestion and we will update the plot with legends.

---

### Official Review · Reviewer_Cjyv · 2022-08-06

**Originality:** Good
**Technical Quality:** Good
**Clarity Of Presentation:** Very Good
**Impact:** 3

**Recommendation:**

Weak Accept: I recommend accepting the paper, but will not argue for my recommendation if the majority of other reviewers have a different opinion.

**Summary:**

The authors introduce a sim2real policy transfer pipeline that learns grasping of rigid objects via an Allegro hand attach to an xArm robot in simulation and transfers the policy learned via reinforcement learning or behavioral cloning to the real robot.
To accomplish the sim2real transfer, the observed point clouds are augmented by an "imagined" point cloud that synthesizes the occluded parts of the robot hand based on proprioceptive state measurements. Furthermore, a contact-based reward function is introduced during training in simulation that encourages contact between the thumb and at least two fingers of the robot hand and the object being grasped. Experiments are conducted on a subset of the YCB objects, as well as as on bottles unseen during training.

**Issues:**

The success metrics between sim and real differ - in simulation a goal position must be reached, but on the real world experiments it suffices to lift up the object a certain height. I do not understand why it is "inconvenient" (l. 190) to measure distance to a target location. This makes the real-world experiments much easier to accomplish.

Why is the observation augmented by a one-hot encoding to indicate whether the point is real or imagined (line 176)? Has this made any difference in the training process?

The caption of Table 1 needs to be updated, the objects types bottle and can do not appear in the upper and bottom two rows.
The caption of Table 2 also does not fit to the table but talks about two figures that are not there.

**Quality Of The Limitations Section:**

Limitations are addressed clearly

**Reviewer Expertise:**

3: The reviewer is fairly confident that the evaluation is correct

**Robotics Focus:**

Relevant but unlikely to deploy to hardware in near future

**Strengths And Weaknesses:**

# Strengths:
The proposed method is technically sound and well motivated. The introduced contact reward function is shown to have a significant impact on the training performance.

The authors provide a useful discussion of limitations of the current work.

# Weaknesses
The paper is focused on grasping, not "dexterous manipulation". The objects are very similar to each other where the robot motion to grasp it does not require any meaningful variation to grasp even the unseen objects, since they all have roughly cylindrical shape. The robot hand essentially just needs to move to the object and then close itself to grasp it. As such, the work does not sufficiently demonstrate its applicability to truly novel objects.

The experiments are lacking metrics for the stability of the grasp that results from these motions. Judging by the supplied video recordings, the hand often does not firmly hold the object in place, so some quantification of robustness to external disturbances is needed.

**Summary Of Recommendation:**

The contributions to include imagined point clouds in the observations and to add contact information to the reward function are promising, but more evaluation is necessary to verify the robustness of the sim2real transfer. Given the simplistic experiments where the objects have very similar shapes and the lack of comparison against other dexterous grasping works, the sim2real performance is difficult to assess.

Edit: Updated my review to weak accept in response to the authors (see explanation below).

---

> ### Author Response · Authors · 2022-08-27
> **Response to Reviewer Cjyv**
>
> We thank the reviewer for their thoughtful comments. We address individual comments in the following.
>
> **Q**: The paper is focused on grasping, not "dexterous manipulation".
>
> **A**: Thank you for the feedback. We indeed have focused mainly on grasping with a dexterous robot hand in the original submission. However, our approach provides a framework that can be generalized to other manipulation tasks. To validate this, we additionally provide a **Door Opening** task here. We have provided new experiment results for the door opening task and uploaded the **policy execution videos** to our [project page](https://dexpc.github.io/). This door opening task requires the robot hand to first rotate the lever to unlock the door latch and then pull the lever in a circular motion to open the door. This task requires delicate coordination between hand wrist motion and finger motion, thus we consider it a challenging dexterous manipulation task. Here we show the evaluation results of door opening tasks in both simulation and real world. We randomized the position of the door relative to the robot base for each trail in both sim and real. The task is considered a success if the door is opened to at least 45 degrees.
>
> **Q**: The robot hand essentially just needs to move to the object and then close itself to grasp it. As such, the work does not sufficiently demonstrate its applicability to truly novel objects.
>
> **A**: We perform new real-robot experiments to increase object diversity. We not only include in-domain objects in the same category but also **out of training domain** objects in different categories. The new real-world experiments consist of 3 categories, which incorporate 26 objects: **10 bottles, 6 cans, and 10 other objects mixed with multiple categories**, shown in [this anonymous link](https://drive.google.com/uc?id=1JI3XXIkTMqjbTg5exqni7E-sxpuCq-32).  These objects require different grasp patterns with diverse shapes. We believe that the new experiments better show the generalizability of our method to novel objects.
>
> |Test Category |Single Obj. Training|Multi Obj. Training|
> |-------|------|------|
> | Bottle Category | $0.75 \pm 0.06$ | $\mathbf{0.87 \pm 0.03}$ |
> | Can Category | $0.75 \pm 0,08$ | $\mathbf{0.83 \pm 0.13}$ |
> | Mixed Category | N/A | $\mathbf{0.73 \pm 0.12}$ |
>
>
> **Q**: The experiments are lacking metrics for the stability of the grasp that results from these motions. Some quantification of robustness to external disturbances is needed.
>
> **A**: We appreciate your constructive suggestions. We add one more experiment and a subsection to discuss the stability analysis of policy. To summarize, we add two perturbations after the object is grasped and lifted: (i) shaking the grasped object with robot arm motion (ii) perturbing the grasped object with external force. We have perform new experiments for the stability test and uploaded the **stability test videos** to our [project page](https://dexpc.github.io/). The success rate before and after two perturbations are shown as follow:
>
> | |Bottle Category|Can Category|
> |----------|--------|------|
> | Before Perturbations | $0.89$ | $0.78$ |
> | After Perturbations | $0.78$ | $0.78$ |
>
>
> **Q**: The success metrics between sim and real differ - in simulation a goal position must be reached, but in real world experiments it suffices to lift up the object a certain height.
>
> **A**: Thanks for the suggestion. In the real world, it is hard to define and measure with a 3D virtual target pose in the air, thus we reduced the problem to lift in our submission. To incorporate the reviewer’s suggestion, we **update our evaluation metric as**: The XY position of the object should be within 5cm from the target position and the object should be lifted at least 15cm. The new metric also considers the horizontal distance, which better captures the task performance. The new real-world experimental results are also performed based on this new metric.
>
> **Q**: Why is the observation augmented by a one-hot encoding to indicate whether the point is real or imagined (line 176)?
>
> **A**: The one-hot encoding indicates whether the point is from the robot hand. Giving such “identities” to the points helps the PointNet encoder to reason about the distance and spatial relation between the hand and the objects, instead of treating every point the same. This provides useful cues for approaching and manipulating objects. To further support our argument, we provide additional experiments here to show the learning curve with or without one-hot encoding on training ShapeNet bottle categories. The learning curves are shown in [this link](https://drive.google.com/uc?id=1YcB3NW3CyMaf7VsYjCuCCXapl0lyR_ku)
>
> **Q**: The caption of Table 1 needs to be updated, the object types bottle and can not appear in the upper and bottom two rows.
>
> **A**: Thank you for pointing these out, we will rephrase the caption of Table 1 and 2 in the revised paper.

---

### Meta-Review · Area_Chair_E46H · 2022-08-09

**Recommendation:** Accept (Poster)
**Confidence:** 4

**Metareview:**

This paper proposes a sim-to-real approach for dexterous manipulation. It trains a policy for a multi-finger robotic hand in simulation, to grasp rigid objects and transfer the policy to the real-world. The proposed method combines point cloud augmentation based on proprioceptive information and reward shaping for the successful zero-shot sim-to-real transfer. The method is validated using a real Allegro hand to grasp new objects of the same category as in training.

While most of the reviews agree that the method seems technically sound and interesting, they brought up three important areas to improve:
1) There is no baseline which the paper compare the proposed algorithm with, even though there are a lot of related work.
2) The objects to grasp in the evaluations are limited: four objects with similar cylinder-like shapes, which cannot sufficiently evaluate the advantage and limitations of the proposed method.
3) The paper focuses on simpler grasping tasks, instead of the claimed "Dexterous Manipulation" in the title.

During the reviewer discussion, all reviewers agreed that the additional experiments had further strengthened the submission. However, the new door opening and lifting cans or bottles still require similar hand motions to achieve the defined tasks. So it is still questionable whether the method exhibits dexterous manipulation rather than grasping. Reviewers suggest that the paper replaces the term "Dexterous Manipulation" in the title to "Multi-fingered Grasping". Nevertheless, reviewers agree that the technical contributions are interesting and the experiments have been improved significantly. For the above reasons, we recommend accepting this paper.

**Best Paper Nomination:**

No

---

> ### Author Response · Authors · 2022-08-27
> **Response to Meta Review**
>
> We’d like to thank the area chair and reviewers for their detailed feedback. We will address questions to each reviewer in individual replies.
>
> ### Baselines
> We agree it is essential to compare with previous methods. Since there is no other work that utilizes point cloud RL for dexterous grasping, we will compare with more widely-used grasp proposal baselines. In the following text, we will first describe the baseline method and then show experimental results and analysis with the baseline approach.
>
> We choose the widely-used EigenGrasp [1] as the grasp representation. Given an object mesh model, we use the well-known GraspIt [2] to search valid grasp for AllegroHand. Then, we use the RRTConnect [3] motion planner implemented in OMPL [4] to plan a joint trajectory to the pre-grasp pose and use a screw motion controller to move the robot from pre-grasp pose to the grasp pose. Finally, we close all fingers in a direction specially designed for Allegro Hand and lift the object to the target. Note that different from our approach, the baseline method requires **complete object model** to search for grasps and **ground-truth object pose** to align the grasp pose in the robot space.
>
> To evaluate the performance of baseline on novel objects, we first build a grasp database on a ShapeNet bottle and can categories using GraspIt. Given the sensory data of a new object, we search for the most similar objects in the dataset and use the query grasps for the novel object. Here we compare the performance of our method with baselines in the real-world.
>
> The success rate comparison is listed in the following table. Note that we include more objects for testing. The bottle category includes 10 bottles and the can category includes 6 cans for testing as shown in [this anonymous link](https://drive.google.com/uc?id=1JI3XXIkTMqjbTg5exqni7E-sxpuCq-32). The oracle means that ground-truth object model and pose are used.
>
> |Method |Bottle Category|Can Category|
> |----------|--------|------|
> | EigenGrasp (oracle) | $0.66$ | $0.50$ |
> | EigenGrasp | $0.45$ | $0.41$ |
> | Ours | $\mathbf{0.87}$ | $\mathbf{0.83}$ |
>
> ### Number of Object in Evaluation
> We perform new real-robot experiments to increase object diversity. We not only include in-domain objects in the same category but also **out of training domain** objects in different categories. The new real-world experiments consist of 3 categories, which incorporate 26 objects: **10 bottles, 6 cans, and 10 other objects mixed with multiple categories**, shown in [this anonymous link](https://drive.google.com/uc?id=1JI3XXIkTMqjbTg5exqni7E-sxpuCq-32).  These objects require different grasp patterns with diverse shapes. We believe that the new experiments better show the generalizability of our method to novel objects.
>
> The updated version of Table 3 in the original submission is as follows:
>
> |Test Category |Single Obj. Training|Multi Obj. Training|
> |----------|--------|------|
> | Bottle Category | $0.75 \pm 0.06$ | $\mathbf{0.87 \pm 0.03}$ |
> | Can Category | $0.75 \pm 0,08$ | $\mathbf{0.83 \pm 0.13}$ |
> | Mixed Category | N/A | $\mathbf{0.73 \pm 0.12}$ |
>
> We have also provided the videos of the policy deployment on these objects on our [project page](https://dexpc.github.io/).
>
> ### More Dexterous Manipulation Task than Grasping
> Thank you for the feedback. We indeed have focused mainly on grasping with a dexterous robot hand in the original submission. However, our approach provides a framework that can be generalized to other manipulation tasks. To validate this, we additionally provide a **Door Opening** task here. We have provided new experiment results for the door opening task and uploaded the **policy execution videos** in our [project page](https://dexpc.github.io/). This door opening task requires the robot hand to first rotate the lever to unlock the door latch and then pull the lever in a circular motion to open the door. This task requires delicate coordination between hand wrist motion and finger motion, thus we consider it a challenging dexterous manipulation task. Here we show the evaluation results of door opening tasks in both simulation and the real world. We randomized the position of the door relative to the robot base for each trail in both sim and real. The task is considered a success if the door is opened to at least 45 degrees. Here we show the success rate of our method on door opening tasks in both the simulator and real world.
>
> | | Success Rate
> |----------|--------|
> | Simulator | $0.92 \pm 0.06$ |
> | Real World | $0.65 \pm 0.05$ |
>
>
> [1] Dimensionality reduction for hand-independent dexterous robotic grasping
>
> [2] GraspIt!: A Versatile Simulator for Grasp Analysis
>
> [3] RRT-Connect: An Efficient Approach to Single-Query Path Planning
>
> [4] The Open Motion Planning Library